# LEARNING THE SPECTROGRAM TEMPORAL RESOLUTION FOR AUDIO CLASSIFICATION

## ABSTRACT

The audio spectrogram is a time-frequency representation that has been widely used for audio classification. The temporal resolution of a spectrogram depends on hop size. Previous works generally assume the hop size should be a constant value such as ten milliseconds. However, a fixed hop size or resolution is not always optimal for different types of sound. This paper proposes a novel method, DiffRes, that enables differentiable temporal resolution learning to improve the performance of audio classification models. Given a spectrogram calculated with a fixed hop size, DiffRes merges non-essential time frames while preserving important frames. DiffRes acts as a "drop-in" module between an audio spectrogram and a classifier, and can be jointly optimized with the classification task. We evaluate DiffRes on the mel-spectrogram, followed by state-of-the-art classifier backbones, and apply it to five different subtasks. Compared with using the fixed-resolution mel-spectrogram, the DiffRes-based method can achieve the same or better classification accuracy with at least 25% fewer temporal dimensions on the feature level, which alleviates the computational cost at the same time. Starting from a high-temporal-resolution spectrogram such as one-millisecond hop size, we show that DiffRes can improve classification accuracy with the same computational complexity.

## 1 INTRODUCTION

Audio classification refers to a series of tasks that assign labels to an audio clip. Those tasks include audio tagging (Kong et al., 2020), speech keyword classfication (Kim et al., 2021), and music genres classification (Castellon et al., 2021). The input to an audio classification system is usually a one-dimensional audio waveform, which can be represented by discrete samples. Although there are methods using time-domain samples as features (Kong et al., 2020; Luo & Mesgarani, 2018; Lee et al., 2017), the majority of studies on audio classification convert the waveform into a spectrogram as the input feature (Gong et al., 2021b;a). Spectrogram is usually calculated by the Fourier transform (Champeney & Champeney, 1987), which is applied in short waveform chunks multiplied by a windowing function, resulting in a two-dimensional time-frequency representation. According to the Gabor's uncertainty principle (Gabor, 1946), there is always a trade-off between time and frequency resolutions. To achieve desired resolution on the temporal dimension, it is a common practice (Kong et al., 2021a; Liu et al., 2022a) to apply a fixed hop size between windows to capture the dynamics between adjacent frames. With the fixed hop size, the spectrogram has a fixed temporal resolution, which we will refer to simply as resolution.

Using a fixed resolution is not necessarily optimal for an audio classification model. Intuitively, the resolution should depend on the temporal pattern: fast-changing signals are supposed to have high resolution, while relatively steady signals or blank signals may not need the same high resolution for the best accuracy (Huzaifah, 2017). For example, Figure 1 shows that by increasing resolution, more details appear in the spectrogram of *Alarm Clock* while the pattern of *Siren* stays mostly the same. This indicates the finer details in high-resolution *Siren* may not essentially contribute to the classification accuracy. There are plenty of studies on learning a suitable frequency resolution with a similar spirit (Stevens et al., 1937; Sainath et al., 2013; Ravanelli & Bengio, 2018b; Zeghidour et al., 2021), but learning temporal resolution is still under-explored. Most previous studies focus on investigating the effect of different temporal resolutions (Kekre et al., 2012; Huzaifah, 2017; Ilyashenko et al., 2019; Liu et al., 2022c). Huzaifah (2017) observe the optimal temporal resolution for audio classifi-

cation is class dependent. Ferraro et al. (2021) experiment on music tagging with coarse-resolution spectrograms, and observes a similar performance can be maintained while being much faster to compute. Kazakos et al. (2021) propose a two-stream architecture that process both fine-grained and coarse-resolution spectrogram and shows the state-of-the-art result on VGG-Sound (Chen et al., 2020). Recently, Liu et al. (2022c) propose a spectrogram-pooling-based module that can improve classification efficiency with negligible performance degradation. In addition, our pilot study shows the optimal resolution is not the same for different types of sound (see Figure 10a in Appendix A.3). The pilot study shows that when increasing resolution, the improvement on different types of sound is not consistent, in which some of them even degrade with a higher resolution. This motivates us to design a method that can learn the optimal resolution.

Besides, the potential of the high-resolution spectrogram, e.g., with one milliseconds (ms) hop size, is still unclear. Some popular choices of hop size including 10 ms (Böck et al., 2012; Kong et al., 2020; Gong et al., 2021a) and 12.5 ms (Shen et al., 2018; Rybakov et al., 2022). Previous studies (Kong et al., 2020; Ferraro et al., 2021) show classification performance can be steadily improved with the increase of resolution. One remaining question is: Can even finer resolution improve the performance? We conduct a pilot study for this question on a limited-vocabulary speech recognition task with hop sizes smaller than 10 ms (see Figure 10b in Appendix A.3). We noticed that accuracy can still be improved with smaller hop size, at a cost of increased computational complexity. This indicates there is still useful information in the higher temporal resolution.

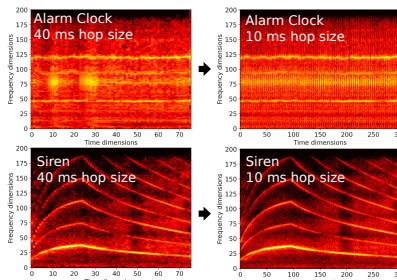

**Figure 1:** The spectrogram of *Alarm Clock* and *Siren* sound with 40 ms and 10 ms hop sizes. All with a 25 ms window size. The pattern of *Siren*, which is relatively stable, does not change significantly using a smaller hop size (i.e., larger temporal resolution), while *Alarm Clock* is the opposite.

In this work, we believe that we are the first to demonstrate learning temporal resolution on the audio spectrogram. We show that learning temporal resolution leads to efficiency and accuracy improvements over the fixed-resolution spectrogram. We propose a lightweight algorithm, DiffRes, that makes spectrogram resolution differentiable during model optimization. DiffRes can be used as a "drop-in" module after spectrogram calculation and optimized jointly with the downstream task. For the optimization of DiffRes, we propose a loss function, guide loss, to inform the model of the low importance of empty frames formed by SpecAug (Park et al., 2019). The output of DiffRes is a time-frequency representation with varying resolution, which is achieved by adaptively merging the time steps of a fixed-resolution spectrogram. The adaptive temporal resolution alleviates the spectrogram temporal redundancy and can speed up computation during training and inference. We perform experiments on five different audio tasks, including the largest audio dataset AudioSet (Gemmeke et al., 2017). DiffRes shows clear improvements on all tasks over the fixed-resolution mel-spectrogram baseline and other learnable front-ends (Zeghidour et al., 2021; Ravanelli & Bengio, 2018b; Zeghidour et al., 2018). Compared with the fixed-resolution spectrogram, we show that using DiffRes can achieve a temporal dimension reduction of at least 25% with the same or better audio classification accuracy. On high-resolution spectrogram, we also show that DiffRes can improve classifier performance without increasing the feature temporal dimensions. Our code is publicly available [1].

## 2 LEARNING TEMPORAL RESOLUTION WITH DIFFRES

We provide an overview of DiffRes-based audio classification in Section 2.1. We introduce the detailed formulation and the optimization of DiffRes in Section 2.2.1, 2.2.2, and 2.3.

### 2.1 OVERVIEW

Let $\boldsymbol{x} \in \mathbb{R}^L$ denote a one-dimensional audio time waveform, where $L$ is the number of audio samples. An audio classification system can be divided into a feature extraction stage and a classification stage. In the feature extraction stage, the audio waveform will be processed by a function $\mathcal{Q}_{l,h} : \mathbb{R}^L \rightarrow \mathbb{R}^{F \times T}$, which maps the time waveform into a two-dimensional time-frequency repre-

---

[1]https://anonymous.4open.science/r/diffres-8F22

**Figure 2:** Audio classification with DiffRes and mel-spectrogram. Green blocks contain learnable parameters. DiffRes is a "drop-in" module between spectrogram calculation and the downstream task.

sentation $\boldsymbol{X}$, such as a mel-spectrogram, where $\boldsymbol{X}_{:,\tau} = (\boldsymbol{X}_{1,\tau}, ..., \boldsymbol{X}_{F,\tau})$ is the $\tau$-th frame. Here, $T$ and $F$ stand for the time and frequency dimensions of the extracted representation. We also refer to the representation along the temporal dimensions as frames. We use $l$ and $h$ to denote window length and hop size, respectively. Usually $T \propto \frac{L}{h}$. We define the temporal resolution $\frac{1}{h}$ by frame per second (FPS), which denotes the number of frames in one second. In the classification stage, $\boldsymbol{X}$ will be processed by a classification model $\mathcal{G}_\theta$ parameterized by $\theta$. The output of $\mathcal{G}_\theta$ is the label predictions $\hat{\boldsymbol{y}}$, in which $\hat{\boldsymbol{y}}_i$ denotes the probability of class $i$. Given the paired training data $(\boldsymbol{x}, \boldsymbol{y}) \in \mathbb{D}$, where $\boldsymbol{y}$ denotes the one-hot vector for ground-truth labels, the optimization of the classification system can be formulated as

$$\arg\min_\theta \mathbf{E}_{(\boldsymbol{x},\boldsymbol{y})\sim\mathbb{D}} \mathcal{L}(\mathcal{G}_\theta(\boldsymbol{X}), \boldsymbol{y}), \qquad (1)$$

where $\mathcal{L}$ is a loss function such as cross entropy (De Boer et al., 2005). Figure 2 show an overview of performing classification with DiffRes. DiffRes is a "drop-in" module between $\boldsymbol{X}$ and $\mathcal{G}_\theta$ focusing on learning the optimal temporal resolution with a learnable function $\mathcal{F}_\phi : \mathbb{R}^{F \times T} \rightarrow \mathbb{R}^{F \times t}$, where $t$ is the parameter denoting the target output time dimensions of DiffRes, and $\phi$ is the learnable parameters. DiffRes formulates $\mathcal{F}_\phi$ with two steps: i) estimating the importance of each time frame with a learnable model $\mathcal{H}_\phi$: $\boldsymbol{X} \rightarrow \boldsymbol{s}$, where $\boldsymbol{s}$ is a $1 \times T$ shape row vector; and ii) warping frames based on a frame warping algorithm, the warping is performed along a single direction on the temporal dimension. We introduce the details of these two steps in Section 2.2.1 and Section 2.2.2. We define the *dimension reduction rate* $\delta$ of DiffRes by $\delta = (T - t)/T$. Usually, $\delta \leq 1$ and $t \leq T$ because the temporal resolution of the DiffRes output is either coarser or equal to that of $\boldsymbol{X}$. Given the same $T$, a larger $\delta$ means fewer temporal dimensions $t$ in the output of DiffRes, and usually less computation is needed for $\mathcal{G}_\theta$. Similar to Equation 1, $\mathcal{F}_\phi$ can be jointly optimized with $\mathcal{G}_\theta$ by

$$\arg\min_{\theta,\phi} \mathbf{E}_{(\boldsymbol{x},\boldsymbol{y})\sim\mathbb{D}} \mathcal{L}(\mathcal{G}_\theta(\mathcal{F}_\phi(\boldsymbol{X})), \boldsymbol{y}). \qquad (2)$$

## 2.2 DIFFERENTIABLE TEMPORAL RESOLUTION MODELING

Figure 2 illustrates adaptive temporal resolution learning with DiffRes. We introduce frame importance estimation in Section 2.2.1, and introduce the warp matrix construction function and frame warping function in Section 2.2.2. Figure 4 shows an example of DiffRes on the mel-spectrogram.

### 2.2.1 FRAME IMPORTANCE ESTIMATION

We design a frame importance estimation module $\mathcal{H}_\phi$ to decide the proportion of each frame that needs to be kept in the output, which is similar to the sample weighting operation (Zhang & Pfister, 2021) in previous studies. The frame importance estimation module will output a row vector $\boldsymbol{s}'$ with shape $1 \times T$, where the element $\boldsymbol{s}'_\tau$ is the importance score of the $\tau$-th time frame $\boldsymbol{X}_{:,\tau}$. The frame importance estimation can be denoted as

$$\boldsymbol{s}' = \sigma(\mathcal{H}_\phi(\boldsymbol{X})), \qquad (3)$$

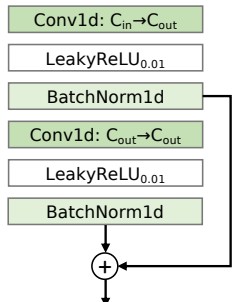

**Figure 3:** ResConv1D

where $\boldsymbol{s}'$ is the row vector of importance scores, and $\sigma$ is the sigmoid function (Han & Moraga, 1995). A higher value in $\boldsymbol{s}'_\tau$ indicates the $\tau$-th frame is important for classification. We apply the sigmoid function to stabilize training by limiting the values in $\boldsymbol{s}'$ between zero and one. We implement $\mathcal{H}_\phi$ with a stack of one-dimensional convolutional neural networks (CNNs) (Fukushima & Miyake, 1982; LeCun et al., 1989). Specifically, $\mathcal{H}_\phi$ is a stack of five one-dimensional convolutional blocks (ResConv1D). We design the ResConv1D block following other CNN based methods (Shu et al., 2021; Liu et al., 2020;

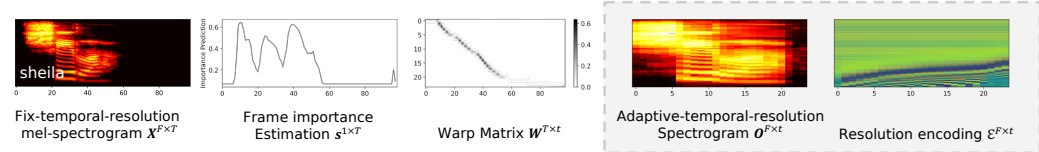

**Figure 4:** Visualizations of the DiffRes using the mel-spectrogram. The part with the shaded background is the input feature of the classifier. For more examples please refer to Figure 12 and 13 in Appendix A.8.

Kong et al., 2021b). As shown in Figure 3, each ResConv1D has two layers of one-dimensional CNN with batch normalization (Ioffe & Szegedy, 2015) and leaky rectified linear unit activation functions (Xu et al., 2015). We apply residual connection (He et al., 2016) for easier training of the deep architecture (Zaeemzadeh et al., 2020). Each CNN layer is zero-padded to ensure the temporal dimension does not change (LeCun et al., 2015). We use exponentially decreasing channel numbers to reduce the computation (see details in Table 10). In the next frame warping step (Section 2.2.2), elements in the importance score will represent the proportion of each input frame that contributes to an output frame. Therefore, we perform rescale operation on $s'$, resulting in an $s$ that satisfies $s \in [0,1]^{1 \times T}$ and $\sum_{k=1}^{T} s_k \leq t$. The rescale operation can be denoted as $\check{s} = \frac{s'}{\sum_{i=1}^{T} s_i'} t$, $\ s = \frac{\check{s}}{\max(\check{s}, 1)}$, where $\check{s}$ is an intermediate variable that may contain elements greater than one, $\max$ denotes the maximum operation. To quantify how active $\mathcal{H}_\phi$ is trying to distinguish between important and less important frames, we also design a measurement, activeness $\rho$, which is calculated by the standard derivation of the non-empty frames, given by

$$\rho = \frac{1}{1-\delta} \sqrt{\frac{\sum_{i \in \mathbb{S}_{\text{active}}} (s_i - \bar{s}_i)^2}{|\mathbb{S}_{\text{active}}|}}, \quad \mathbb{S}_{\text{active}} = \{i \mid \text{E}(\boldsymbol{X}_{:,i}) > \min(\text{E}(\boldsymbol{X}_{:,i})) + \epsilon\}, \quad (4)$$

where $\mathbb{S}_{\text{active}}$ is the set of indices of non-empty frames, $\epsilon$ is a small value, $|\mathbb{S}|$ denotes the size of set $\mathbb{S}$, function $\text{E}(\cdot)$ calculates the root-mean-square energy (Law & Rennie, 2015) of a frame in the spectrogram, and function $\min(\cdot)$ calculates the minimum value within a matrix. We use $\delta$ to unify the value of $\rho$ for easier comparison between different $\delta$ settings. The activeness $\rho$ can be used as an indicator of how DiffRes behaves during training. A higher $\rho$ indicates the model is more active at learning the frame importance. A lower $\rho$ such as zero indicates learning nothing. We will discuss the learning process of DiffRes with $\rho$ in Section 3.3.

### 2.2.2 TEMPORAL FRAME WARPING

We perform temporal frame warping based on $s$ and $X$ to calculate an adaptive temporal resolution representation $O$, which is similar to the idea of generating derived features (Pentreath, 2015). Generally, the temporal frame warping algorithm can be denoted by $\boldsymbol{W} = \alpha(\boldsymbol{s})$ and $\boldsymbol{O} = \beta(\boldsymbol{X}, \boldsymbol{W})$, where $\alpha(\cdot)$ is a function that convert $s$ into a warp matrix $W$ with shape $t \times T$, and $\beta(\cdot)$ is a function that applies $W$ to $X$ to calculate the warpped feature $O$. Elements in $W$ such as $\boldsymbol{W}_{i,j}$ denote the contribution of the $j$-th input frame $\boldsymbol{X}_{:,j}$ to the $i$-th output frame $\boldsymbol{O}_{:,i}$. We will introduce the realization of $\alpha(\cdot)$ and $\beta(\cdot)$ in the following sections. Figure 11 in Appendix A.7 provides an example of the warp matrix construction function.

**Warp matrix construction** Function $\alpha(\cdot)$ calculates the warp matrix $W$ with $s$ by:

$$\boldsymbol{W}_{i,j} = \begin{cases} \boldsymbol{s}_j, & \text{if } i < \sum_{k=1}^{j} \boldsymbol{s}_k \leq i+1 \\ 0, & \text{otherwise} \end{cases}, \quad (5)$$

where we calculate the cumulative sum of $s$ to decide which output frame each input frame will be warped into. The warp matrix $W$ here can be directly used for frame warping function $\beta(\cdot)$. However, we found the DiffRes can learn more actively if the warp matrix satisfies $\sum_{j=1}^{t} \boldsymbol{W}_{j,:} = \mathbf{1}$, which means each output frame is assigned an equal amount of total warp weights on input frames. We further process $W$ to meet this requirement with Algorithm 1 in Appendix A.6. We also provide a vectorized version of Algorithm 1 that can be run efficiently on GPUs (see Appendix A.7).

**Frame warping** Function $\beta(\cdot)$ performs frame warping based on the warp matrix $W$. The $i$-th output frame is calculated with $X$ and the $i$-th row of $W$, given by

$$\boldsymbol{O}_{i,j} = \mathcal{A}((\boldsymbol{X}_{j,:}) \odot (\boldsymbol{W}_{i,:})), \quad (6)$$

where $\mathcal{A} : \mathbb{R}^{1 \times T} \to \mathbb{R}$ stands for the frame aggregation function such as averaging, $O$ is the final output feature with shape $F \times t$.

**Resolution encoding** The final output $O$ does not contain the resolution information at each time step, which is crucial information for the classifier. Since the temporal resolution can be represented with $W$, we construct a resolution encoding with $W$ in parallel with frame warping. Firstly, we construct a positional encoding matrix $E$ with shape $F \times T$, using the similar method described in Vaswani et al. (2017). Each column of $E$ represents a positional encoding of a time step. Then we calculate the resolution encoding by $\mathcal{E} = EW^\top$, where $W^\top$ stands for the transpose of $W$. The shape of the resolution encoding is $F \times t$. Both $\mathcal{E}$ and $O$ are concatenated on the channel dimension as the classifier input feature.

## 2.3 OPTIMIZATION

We propose a guide loss to provide guidance for DiffRes on learning frame importance. Since we do not know the ground truth frame importance, we cannot directly optimize $s$. We introduce $\mathcal{L}_{guide}$ as an inductive bias (Mitchell, 1980) to the system based on the assumption that an empty frame should have a low importance score. Specifically, we propose the guide loss by

$$\mathcal{L}_{guide} = \frac{1}{|\mathbb{S}_{empty}|} \sum_{i \in \mathbb{S}_{empty}} (\frac{s_i}{1-\delta} - \lambda)^+, \quad \mathbb{S}_{empty} = \{i \mid i \notin \mathbb{S}_{active} \text{ and } i \in \{1, 2, ..., T\}\}, \quad (7)$$

where $\mathbb{S}_{empty}$ is a set of time indexes that have low energy, and $\lambda$ is a constant threshold. Given that the output of DiffRes has fewer temporal dimensions than $X$, the DiffRes layer forms an information bottleneck (Tishby et al., 2000; Shwartz-Ziv & Tishby, 2017) that encourages DiffRes to assign a higher score to important frames. We analyze the information bottleneck effect of DiffRes in Section 3.3. The parameter $\lambda$ is a threshold for the guide loss to take effect. This threshold can alleviate the modeling bias toward energy. For example, if $\lambda = 0$, the importance scores of empty frames are strongly regularized, and the model will also tend to predict low importance scores for lower energy frames, which may contain useful information. $\mathcal{L}_{bce}$ is the standard binary cross entropy loss function (Shannon, 2001) for classification, given by Equation 8, where $\hat{y}$ is the label prediction and $N$ is the total number of classes.

$$\mathcal{L}_{bce} = \frac{1}{N} \sum_{i=1}^{N} (y_i \log(\hat{y}_i) + (1 - y_i) \log(1 - \hat{y}_i)), \quad (8)$$

The loss function of the DiffRes-based audio classification system includes our proposed guide loss $\mathcal{L}_{guide}$ and the binary cross entropy loss $\mathcal{L}_{bce}$, given by $\mathcal{L} = \mathcal{L}_{bce} + \mathcal{L}_{guide}$.

## 3 EXPERIMENTS

We focus on evaluating DiffRes on the mel-spectrogram, which is one of the most popular features used by state-of-the-art systems (Chen et al., 2022; Gong et al., 2022; Verbitskiy et al., 2022; Koutini et al., 2021). We evaluate DiffRes on five different tasks and datasets (see Table 9 in Appendix A.5), including audio tagging on AudioSet (Gemmeke et al., 2017) and FSD50K (Fonseca et al., 2021), environmental sound classification on ESC50 (Piczak, 2015), limited-vocabulary speech recognition on SpeechCommands (Warden, 2018), and music instrument classification on NSynth (Engel et al., 2017). All the datasets are resampled at a sampling rate of 16 kHz. Following the evaluation protocol in the previous works (Zeghidour et al., 2021; Riad et al., 2021; Kong et al., 2020; Gong et al., 2021b), we report the mean average precision (mAP) as the main evaluation metric on AudioSet and FSD50K, and report classification accuracy (ACC) on other datasets. In all experiments, we use the same architecture as used by Gong et al. (2021b), which is an EfficientNet-B2 (Tan & Le, 2019) with four attention heads (13.6 M parameters). We reload the ImageNet pretrained weights for EfficientNet-B2 in a similar way to Gong et al. (2021a;b). We also provide ablation study on other model architectures in Appendix A.1.2. For the training data, we apply random spec-augmentation (Park et al., 2019) and mixup augmentation (Zhang et al., 2017) following Gong et al. (2021b). All experiments are repeated three times with different seeds to reduce randomness. We also report the standard derivation of the repeated trails along with the averaged result. We train the DiffRes layer with $\lambda = 0.5$ and $\epsilon = 1 \times 10^{-4}$. For the frame aggregation function $\mathcal{A}$ (see Equation 6), we use both the max and mean operations, whose outputs are concatenated with the resolution encoding $\mathcal{E}$ on the channel dimension as the input feature to the classifier. The frame importance estimation module we used in this paper is a stack of five ResConv1D (see Table 10 in

| Task name 100 FPS baseline (%) | Metric | FPS | Change hop size (%) | AvgPool (%) | ConvAvgPool (%) | Proposed (%) |
|---|---|---|---|---|---|---|
| AudioSet tagging 43.7 ± 0.1 | mAP | 25 | 38.6 ± 0.3 | 39.9 ± 0.2 | 40.1 ± 0.2 | **41.7** ± 0.1 |
|  |  | 50 | 41.8 ± 0.2 | 42.4 ± 0.1 | 42.7 ± 0.2 | **43.6** ± 0.1 |
|  |  | 75 | 42.7 ± 0.2 | 43.6 ± 0.0 | 43.5 ± 0.2 | **44.2** ± 0.1$^\dagger$ |
| FSD50K tagging 55.6 ± 0.3 | mAP | 25 | 48.9 ± 0.4 | 51.4 ± 0.3 | 49.2 ± 0.4 | **56.9** ± 0.2$^\dagger$ |
|  |  | 50 | 53.3 ± 0.4 | 54.5 ± 0.4 | 52.2 ± 0.8 | **57.2** ± 0.2$^\dagger$ |
|  |  | 75 | 54.8 ± 0.4 | 55.3 ± 0.3 | 54.4 ± 0.2 | **57.1** ± 0.4$^\dagger$ |
| Environmental sound 85.2 ± 0.5 | ACC | 25 | 74.6 ± 0.6 | 75.6 ± 0.3 | 72.4 ± 1.2 | **82.9** ± 0.5 |
|  |  | 50 | 82.4 ± 0.5 | 83.2 ± 0.3 | 77.3 ± 0.8 | **85.5** ± 0.4$^\dagger$ |
|  |  | 75 | 84.9 ± 0.3 | 85.2 ± 0.4$^\dagger$ | 81.8 ± 0.6 | **86.8** ± 0.3$^\dagger$ |
| Speech recognition 97.2 ± 0.1 | ACC | 25 | 93.5 ± 0.1 | 94.9 ± 0.4 | **95.8** ± 0.3 | 95.0 ± 0.3 |
|  |  | 50 | 96.1 ± 0.1 | 96.0 ± 0.2 | 96.0 ± 0.1 | **96.7** ± 0.2 |
|  |  | 75 | 96.8 ± 0.2 | 96.9 ± 0.1 | 97.0 ± 0.1 | **97.2** ± 0.0$^\dagger$ |
| Music instrument 79.9 ± 0.2 | ACC | 25 | 79.7 ± 0.2 | 78.3 ± 0.7 | 78.0 ± 0.5 | **80.5** ± 0.2$^\dagger$ |
|  |  | 50 | 79.9 ± 0.0$^\dagger$ | 79.5 ± 0.3 | 79.4 ± 0.3 | **81.0** ± 0.5$^\dagger$ |
|  |  | 75 | 79.8 ± 0.2 | 79.6 ± 0.3 | 79.7 ± 0.4 | **80.8** ± 0.2$^\dagger$ |

**Table 1:** Comparison of different temporal dimension reduction methods. The numbers under the task name show the baseline performance. Baseline methods use fix-temporal-resolution mel-spectrogram with 10 ms hop size. Numbers with † mean better or comparable performance compared with the 100 FPS baseline.

Appendix A.5) with a total 82 371 of parameters. We calculate the mel-spectrogram with a Hanning window, 25 ms window length, 10 ms hop size, and 128 mel-filterbanks by default. We list our detailed hyperparameters setting in Table 8 (Appendix A.5). We also encourage readers to read through Figure 12 and Figure 13 in Appendix A.8 for examples of DiffRes on the mel-spectrogram.

## 3.1 ADAPTIVELY COMPRESS THE TEMPORAL DIMENSION

Compression of mel-spectrogram temporal dimension can lead to a considerable speed up on training and inference (see Figure 5), thus it is a valuable topic for efficient classification (Huang & Leanos, 2018) and on-device application (Choi et al., 2022). The experiments in this section aim to evaluate the effectiveness of DiffRes in compressing temporal dimensions and maintaining precision. We compare DiffRes with three temporal dimension reduction methods: i) *Change hop size* (CHSize) reduces the temporal dimension by enlarging the hop size. The output of CHSize has a fixed resolution and may lose information between output frames.; ii) *AvgPool* is a method that performs average pooling on a 100 FPS spectrogram to reduce the temporal dimensions. AvgPool also has a fixed resolution, but it can aggregate information between output frames by pooling; iii) *ConvAvgPool* is the setting that the 100 FPS mel-spectrogram will be processed by a stack of ResConv1D (mentioned in Section 2.2.1), followed by an average pooling for dimension reduction. ConvAvgPool has a total of 493 824 parameters and we provide the detailed structure in Table 10 (Appendix A.5). Based on a learnable network, ConvAvgPool has the potential of learning more suitable features and temporal resolution implicitly.

**Baseline comparisons** Table 1 shows our experimental result. The baseline of this experiment is performed on mel-spectrogram without temporal compression (i.e., 100 FPS) and the baseline result is shown under each task name. When reducing 25% of the temporal dimension (i.e., 75 FPS), the proposed method can even considerably improve the baseline performance on most datasets, except on speech recognition tasks where we maintain the same performance. We assume the improvement comes from the data augmentation effect of DiffRes, which means divergent temporal compression on the same data at different training steps. With a 50 FPS, four out of five datasets can maintain comparable performance. With only 25 FPS, the proposed method can still improve the FSD50K tagging and music instrument classification tasks, which indicates the high temporal redundancy in these datasets. Our proposed method also significantly outperforms other temporal dimension reduction baselines. With fixed resolution and fewer FPS, the performance of CHSize degrades more notably. AvgPool can outperform CHSize by aggregating more information between output frames. Although ConvAvgPool has an extra learnable neural network, it does not show significant improvements compared with AvgPool. ConvAvgPool even has an inferior performance on FSD50K and environmental sound classification tasks. This indicates employing a neural network for feature

learning is not always beneficial. Also, by comparison, the interpretability of our proposed method is much better than ConvAvgPool (see Figure 9).

**On variable-length audio data** Note that the proposed method even improves the mAP performance by 1.3% with only 25 FPS on the FSD50K dataset. This is because the audio clip durations in the FSD50K have a high variance (see Table 9). In previous studies (Gong et al., 2021a;b; Kong et al., 2020), a common practice is padding the audio data into the same duration in batched training and inference, which introduces a considerable amount of temporal redundancy in the data with a significantly slower speed. By comparison, DiffRes can unify the audio feature shape regardless of their durations. Model optimization becomes more efficient with DiffRes. As a result, the proposed method can maintain an mAP of $55.6 \pm 0.2$ on the FSD50K, which is comparable to the baseline, with only 15 FPS and 28% of the original training time. This result shows that DiffRes provides a new mind map for future work on classifying variable-length audio files.

## 3.2 LEARNING WITH SMALLER HOP SIZE

Previous studies have observed that a higher resolution spectrogram can improve audio classification accuracy (Kong et al., 2020; Ferraro et al., 2021). However, a hop size smaller than 10 ms has not been widely explored. This is partly because the computation becomes heavier for a smaller hop size. For example, with 1 ms hop size (i.e., 1000 FPS), the time and space complexity for an EfficientNet classifier will be 10 times heavier than with a common 10 ms hop size. Since DiffRes can control the temporal dimension size, namely FPS, working on a small hop size spectrogram becomes computationally friendly. Table 2 shows model performance can be considerably improved with smaller hop sizes. AudioSet and environment sound dataset achieve the best performance on 6 ms and 1 ms hop size, and other tasks benefit most from 3 ms hop sizes. In later experiments, we will use these best hop size settings on each dataset.

| Hop size | 10 ms | 6 ms | 3 ms | 1 ms |
|---|---|---|---|---|
| FPS (Dimension reduction rate) | 100 (0%) | 100 (40%) | 100 (70%) | 100 (90%) |
| AudioSet tagging | $43.7 \pm 0.1$ | $\mathbf{44.1} \pm 0.1$ | $43.8 \pm 0.0$ | $43.7 \pm 0.1$ |
| Environmental Sound Classification | $85.2 \pm 0.4$ | $87.2 \pm 0.3$ | $88.0 \pm 0.6$ | $\mathbf{88.4} \pm 0.5$ |
| Speech recognition | $97.2 \pm 0.1$ | $97.6 \pm 0.0$ | $\mathbf{97.9} \pm 0.1$ | $97.8 \pm 0.1$ |
| Music instrument | $79.9 \pm 0.2$ | $81.3 \pm 0.3$ | $\mathbf{81.8} \pm 0.2$ | $80.6 \pm 0.4$ |
| Average | $76.5 \pm 0.2$ | $77.6 \pm 0.2$ | $\mathbf{77.9} \pm 0.1$ | $77.5 \pm 0.2$ |

**Table 2:** Learning with high temporal resolution spectrograms. FPS is controlled at 100, so the computational complexity of the classifier is the same in all hop-size settings. Results are reported in the percentage format.

**Comparing with other learnable front-ends** The DiffRes is learnable, so the Mel+DiffRes setting as a whole can be viewed as a learnable front-end. Table 3 compares our proposed method with SOTA learnable front-ends, our best setting is denoted as Mel+DiffRes (Best), which achieves the best result on all datasets. For a fair comparison, we control the experiment setup to be consistent with Zeghidour et al. (2021) in Mel+DiffRes. Specifically, we change the backbone to EfficientNet-B0 (5.3 M parameters) without ImageNet pretraining. We also remove spec-augment and mixup, except in AudioSet, and change our Mel bins from 128 to 40, except in the AudioSet experiment where we change to 64. The result shows Mel+DiffRes can outperform SOTA learnable front-end (Zeghidour et al., 2021; Ravanelli & Bengio, 2018b; Zeghidour et al., 2018) by a large margin.

| Front-end | Mel | TD-fbank | SincNet | LEAF | Mel+DiffRes | Mel+DiffRes (Best) |
|---|---|---|---|---|---|---|
| Parameters | 0 | 51 k | 256 | 448 | 82 k | 82 k |
| AudioSet tagging | $96.8 \pm 0.1$ | $96.5 \pm 0.1$ | $96.1 \pm 0.0$ | $96.8 \pm 0.1$ | $97.0 \pm 0.0$ | $\mathbf{97.5} \pm 0.0$ |
| Speech recognition | $93.6 \pm 0.3$ | $89.5 \pm 0.4$ | $91.4 \pm 0.4$ | $93.6 \pm 0.3$ | $95.4 \pm 0.2$ | $\mathbf{97.9} \pm 0.1$ |
| Music instrument | $70.7 \pm 0.6$ | $66.3 \pm 0.6$ | $67.4 \pm 0.6$ | $70.2 \pm 0.6$ | $78.5 \pm 0.7$ | $\mathbf{81.8} \pm 0.2$ |
| Average | $87.0 \pm 0.3$ | $84.1 \pm 0.4$ | $85.0 \pm 0.3$ | $86.9 \pm 0.3$ | $90.3 \pm 0.3$ | $\mathbf{92.4} \pm 0.1$ |

**Table 3:** Comparison with SOTA learnable front-ends. All the methods use 100 FPS. Results are reported in the percentage format. Mel+DiffRes controls the experimental settings mentioned in Section 3.2 to be consistent with Mel, TD-fbank, SincNet, and LEAF. Mel+DiffRes (Best) use the best possible settings.

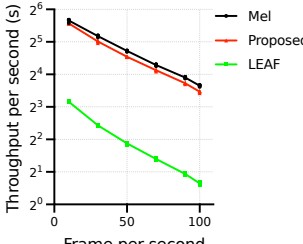
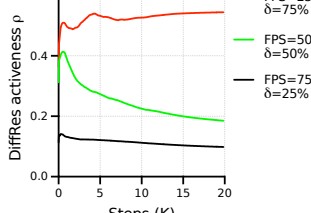
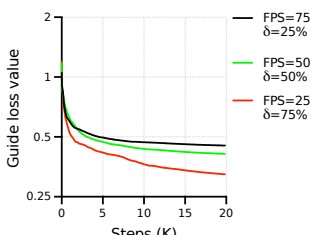

**Figure 5:** Audio throughput in one second. Evaluated on a 2.6 GHz Intel Core i7 CPU.

**Figure 6:** Trajectories of DiffRes learning activeness ($\rho$) on different training steps and FPS settings.

**Figure 7:** The training curve of guide loss ($\mathcal{L}_{guide}$) with different FPS settings.

**Computational cost** We assess the one-second throughput of different front-ends on various FPS settings to compare their computational efficiency. We control the FPS of Mel and LEAF by average pooling. The computation time is measured between inputting waveform and outputting label prediction (with EfficientNet-B2). We use 128 filters in LEAF (Zeghidour et al., 2021) for a fair comparison with 128 mel-filterbanks in Mel and DiffRes. As shown in Figure 5, our proposed DiffRes only introduces marginal computational cost compared with Mel. The state-of-the-art learnable front-end, LEAF, is about four times slower than our proposed method. The majority of the cost in computation in LEAF comes from multiple complex-valued convolutions, which are computed in the time-domain with large kernels (e.g., 400) and a stride of one.

## 3.3 Analysis for the learning of DiffRes

**Learning activeness** DiffRes does not explicitly learn the optimal frame importance score because the ground truth frame importance is not available. Instead, DiffRes is optimized with the guidance of guide loss $\mathcal{L}_{guide}$ (Equation 7), which is a strong assumption we introduced to the model. Figure 6 shows the trajectories of the DiffRes learning activeness (defined in Section 2.2.1) during the optimization with different FPS settings on the speech recognition task in Table 1. According to the final converged value, DiffRes with a smaller FPS tend to be more active at learning frame importance. This is intuitive since smaller FPS leads to more information bottleneck effects (Saxe et al., 2019) in DiffRes. With a 25 FPS, the activeness even keeps increasing with more training steps, indicating the active learning of DiffRes. Figure 7 shows the guide loss curve during training with different FPS settings. Intuitively, when the FPS is small, a model needs to preserve more non-empty frames and fewer empty frames for better accuracy. This assumption is aligned with our experiment result, which shows the model tends to have a lower guide loss with a smaller FPS.

**Data augmentation and regularization effect** As reflected in the curve of $\rho$ and $\mathcal{L}_{guide}$ in Figure 6 and 7, DiffRes is optimized along with the classifier during training. Hence DiffRes may produce different outputs for the same training data at different epochs. This is equivalent to performing data augmentation on the audio data. We suppose this is the main reason for the improved performance shown in Table 1. Also, DiffRes reduces the sparsity of the audio feature by adaptive temporal compression. This is equivalent to performing an implicit regularization (Neyshabur, 2017; Arora et al., 2019) on the feature level, which is beneficial for the system efficiency.

**Ablation studies** We perform the ablation study on AudioSet (Gemmeke et al., 2017), the largest audio dataset by far. The ablation study is performed on the mel-spectrogram with 10 ms hop size and a 75% DiffRes dimension reduction rate (i.e., 25 FPS). Figure 8 shows the value of guide loss, activeness, and mean average precision (mAP) with different training steps and ablation setups. As the red solid line shows, if we remove the guide loss, the DiffRes activeness becomes very high, but the mAP becomes worse. Meanwhile, the value of guide loss increases gradually, which is counter-intuitive because the empty frames should not have high important scores. This indicates guide loss is an effective prior knowledge we can introduce to the system. As the dotted green line shows, if we remove the resolution encoding, the curve of guide loss and activeness almost show no changes, while the mAP degrades 1.15%. This indicates resolution encoding is crucial information for the classifier. If we remove the max aggregation function (see the dotted black line), both the activeness and the mAP have a notable degradation. As shown by the green solid line, if we do not apply Algorithm 1, the DiffRes activeness will converge to a small value, which means DiffRes is not working active enough, indicating Algorithm 1 is crucial for DiffRes to take effects.

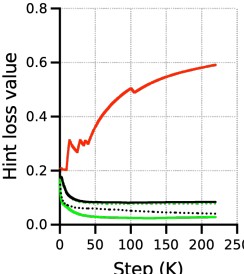 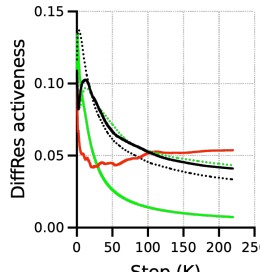 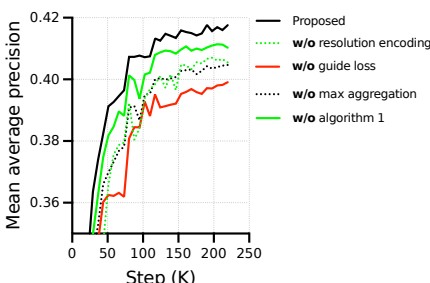

**Figure 8:** The ablation study on AudioSet tagging task with 10 ms hop size and 75% dimension reduction rate. Three figures visualize the guide loss, DiffRes activeness $\rho$, and the mean average precision (mAP), respectively, with different training steps and ablation setups.

For a more detailed analysis of different hyper-parameter values in the guide loss, please refer to Appendix A.1.1. We also provide a weakness analysis of DiffRes in Appendix A.4.

## 4 RELATED WORKS

Neural-network based methods have been successfully applied on audio classification and achieved state-of-the-art performance, such as the pretrained audio neural networks (PANNs) (Kong et al., 2020), pretraining, sampling, labeling, and aggregation based audio tagging (PSLA) (Gong et al., 2021b), and audio spectrogram transformer (AST) (Gong et al., 2021a). We will cover two related topics on audio classification in the following sections.

**Learnable audio front-ends** In recent years, learning acoustic features from waveform using trainable audio front-ends has attracted increasing interest from researchers. Sainath et al. (2013) introduced one of the earliest works that propose to jointly learn the parameter of filter banks with a speech recognition model. Later, SincNet (Ravanelli & Bengio, 2018b) proposes to learn a set of bandpass filters on the waveform and has shown success on speaker recognition (Ravanelli & Bengio, 2018b;a). Most recently, (Zeghidour et al., 2021) proposes to learn bandpass, and low-pass filtering as well as per-channel compression (Wang et al., 2017) simultaneously in the audio front-end and shows consistent improvement in audio classification. Different from existing work on learnable audio front-ends, which mostly focus on the frequency dimension, our objective is learning the optimal temporal resolution. We show that our method can outperform existing audio front-ends for audio classification on both accuracy and computation efficiency (see Table 3 and Figure 5). Note that our proposed method can also be applied after most learnable front-ends (Zeghidour et al., 2021), which will be our future direction.

**Learning feature resolution with neural networks** One recent work on learning feature resolution for audio classification is DiffStride (Riad et al., 2021), which learns stride in convolutional neural network (CNN) in a differentiable way and outperforms previous methods using fixed stride settings. By comparison, DiffStride needs to be applied in each CNN layer and can only learn a single fixed stride setting, while DiffRes is a one-layer lightweight algorithm and can personalize the best temporal resolution for each audio during inference. Recently, Gazneli et al. (2022) proposed to use a stack of one-dimension-CNN blocks to downsample the audio waveform before the audio classification backbone network, e.g., Transformer, which can learn temporal resolution implicitly for audio classification. In contrast, DiffRes can explicitly learn temporal resolution on the feature level with similar interpretability as mel-spectrogram.

## 5 CONCLUSIONS

In this paper, we introduce DiffRes, a "drop-in" differentiable temporal resolution learning module that can be applied between audio spectrogram and downstream tasks. For the training of DiffRes, our proposed guide loss is shown to be beneficial. We demonstrate over a large range of tasks that DiffRes can improve or maintain similar performance with 25% to 75% reduction on temporal dimensions, and can also efficiently utilize the information in high-resolution spectrograms to improve accuracy. In future work, we will move forward to evaluate DiffRes on different kinds of time-frequency representations with more sophisticated frame importance prediction models. Also, we believe the idea of DiffRes can work on other time series data as well for learning optimal temporal resolutions, such as video or seismic data.

## 6 REPRODUCIBILITY STATEMENT

We believe that we have covered the details of our proposed method with figures (see Figures 2 to 13), algorithms (see Algorithm 1 to 3) and mathematical equations (see Equation 2 to 8). As shown in Table 9 and 8, the hyperparameters are provided as exhaustive as we can, both in the main text and the appendix. The realization of the LEAF module we used for throughput comparison is https://github.com/SarthakYadav/leaf-pytorch. We also open-source our code at https://anonymous.4open.science/r/diffres-8F22 to facilitate reproducibility.

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

# A APPENDIX

## A.1 ABLATION STUDIES

We report ablation studies on hyper-parameters and different model architectures in Appendix A.1.1 and A.1.2, respectively. In Appendix A.1.3, we also discuss whether DiffRes only learns to remove silent frames.

### A.1.1 HYPER-PARAMETERS

In this section, we provide ablation studies and discussions on the hyper-parameters in DiffRes, including the threshold $\lambda$ mentioned in Equation 7, dimension reduction rate $\delta$, and the $\epsilon$ used in Equation 4. We choose to conduct the experiment on the SpeechCommands dataset since it has a reasonable amount of data and is computationally friendly on model training compared with large datasets such as AudioSet (Gemmeke et al., 2017). The ablation study result on hyper-parameter is presented in Table 4.

| Metric | Hop size (ms) | $\delta$ | $\lambda$=0.0 | $\lambda$=0.3 | $\lambda$=0.5 | $\lambda$=0.8 | Average | $\prime$ |
|---|---|---|---|---|---|---|---|---|
| Accuracy (%) | 3 | 70% | 98.0 | 97.9 | 98.0 | 98.0 | $98.0 \pm 0.0$ | 97.8 |
| | 1 | 90% | 98.0 | 98.0 | 98.0 | 98.0 | $98.0 \pm 0.0$ | 97.8 |
| | 0.5 | 95% | 97.9 | 97.9 | 97.9 | 98.0 | $97.9 \pm 0.0$ | 97.7 |
| Activeness $\rho$ (%) | 3 | 70% | 32.4 | 29.4 | 28.1 | 30.6 | $30.1 \pm 1.6$ | 20.6 |
| | 1 | 90% | 42.3 | 42.8 | 43.9 | 42.4 | $42.9 \pm 0.6$ | 17.9 |
| | 0.5 | 95% | 43.1 | 41.6 | 44.7 | 40.5 | $42.5 \pm 1.6$ | 8.4 |
| Average importance scores on empty frames (%) | 3 | 70% | 0.2 | 13.2 | 27.2 | 41.0 | $20.4 \pm 17.6$ | 81.1 |
| | 1 | 90% | 0.2 | 4.6 | 11.8 | 30.5 | $11.8 \pm 13.4$ | 91.6 |
| | 0.5 | 95% | 0.2 | 3.0 | 11.0 | 17.2 | $7.8 \pm 7.7$ | 102.3 |
| Guide loss applied | | | ✓ | ✓ | ✓ | ✓ | ✓ | ✗ |

**Table 4:** Ablation study on the SpeechCommands dataset. All the experiments use 100 FPS. The baseline performance is $97.2 \pm 0.1$ with 10 ms hop size and $\delta = 0\%$. We report the accuracy, activeness, and average importance score on empty frames on different hop size, dimension reduction rate $\delta$, guide loss, and threshold $\lambda$ settings. The column "Average" denotes the average result on each metric with four different $\lambda$ values.

**The effect of guide loss.** Table 4 shows that even without guide loss, the model can still improve over the baseline performance ($97.2 \pm 0.1$) using DiffRes. At the same time, applying guide loss can further improve the activeness $\rho$ (see Equation 4) of DiffRes and classification performance. For example, without the guide loss, the $\rho$ with 3 ms, 1 ms, and 0.5 ms hop size are 20.6, 17.9, and 8.4, respectively, while after applying guide loss, the average $\rho$ become 32.6, 45.4, and 45.0, respectively. The improvement on $\rho$ indicates guide loss can encourage the model to better discriminate between the importance of frames. The model classification accuracy can improve by about 0.2% after applying guide loss, which is significant enough for the SpeechCommands dataset. Moreover, without guide loss, the model tends to predict high-importance scores on empty frames, which is also counterintuitive.

**The effect of dimension reduction rate** $\delta$ With the same hop size, a smaller $\delta$ will lead to a larger temporal dimension in the DiffRes output feature, which also leads to heavier computational cost (see Figure 5). Even though a smaller hop size and smaller $\delta$ tend to achieve better performance because finer temporal details can be preserved, in practice, the exact value of $\delta$ still should be determined by the computation limit.

**The effect of guide loss threshold** $\lambda$ As shown in Table 4, we tried different $\lambda$ on different hop sizes. The experimental result shows model accuracy is not sensitive to $\lambda$ thus the value of $\lambda$ usually does not need careful finetuning.

**The effect of the small value** $\epsilon$ We use $\epsilon$ in Equation 4 to control the threshold of deciding whether each frame is active or empty. In practice, we will apply SpecAug (Park et al., 2019) on the spectrogram, in which the empty frames $\mathbb{S}_{empty}$ in Equation 7 will be the masked time steps. To verify $\epsilon$ is not essential for model training, We try to construct $\mathbb{S}_{empty}$ in Equation 7 on training data with five

different $\epsilon$ values between $1 \times 10^{-4}$ and $1 \times 10^{-8}$. Our result shows more than 98% training data have the same $\mathbb{S}_{\text{empty}}$ with different $\epsilon$ values, which indicates $\epsilon$ is not an essential hyper-parameter for model training.

## A.1.2 MODEL ARCHITECTURE

To verify the generality of the proposed approach, we also conduct experiments on two more state-of-the-art architectures, CNN6 and CNN14 (Kong et al., 2020). Experiments are conducted on the SpeechCommands dataset with the same setting as Table 1.

| Hop Size (ms) | $\delta$ | Frames per second | EfficientNet-b2 | CNN6 | CNN14 |
|---|---|---|---|---|---|
| 10 | 0% | 100 | 97.2±0.1 | $96.4 \pm 0.1$ | $97.9 \pm 0.1$ |
| 10 | 25% | 75 | $97.2 \pm 0.0$ | $96.4 \pm 0.0$ | $98.0 \pm 0.0$ |
| 10 | 50% | 50 | $96.7 \pm 0.2$ | $96.1 \pm 0.1$ | $97.7 \pm 0.0$ |
| 10 | 75% | 25 | $95.0 \pm 0.3$ | $95.7 \pm 0.1$ | $97.1 \pm 0.1$ |
| 6 | 40% | 100 | $97.6 \pm 0.0$ | $96.8 \pm 0.0$ | $98.1 \pm 0.0$ |
| 3 | 70% | 100 | $97.9 \pm 0.1$ | $97.2 \pm 0.1$ | $98.1 \pm 0.0$ |
| 1 | 90% | 100 | $97.8 \pm 0.1$ | $97.2 \pm 0.0$ | $98.1 \pm 0.2$ |

**Table 5:** Ablation study on the model architectures. We use $\delta$ to denote the dimension reduction rate. Large $\delta$ indicates less computational cost.

Table 5 shows our ablation study result on different architectures. Three results exhibit a similar trend as Table 1 and Table 2. All three models can maintain a similar or better performance after reducing 25% of the temporal dimensions. With the same number of frames per second, namely the same computational cost, all the models show clear improvement with a smaller hop size. This improvement indicates DiffRes is effective in selecting informative frames across different architectures. We do not experiment with other non-neural architecture because the optimization of DiffRes requires gradient back-propagations (LeCun et al., 2015).

## A.1.3 REMOVE EMPTY FRAME OR SELECT IMPORTANT FRAME?

To study whether DiffRes learns to remove only silent frames, or if it would be also effective when the signal has consistent energy, we design a pitch classification experiment on the NSynth dataset following (Zeghidour et al., 2021). We will refer to this task as NSynth-Pitch. We design the pitch classification task for the following two reasons: (i) The data in NSynth is mostly instrumental sounds, which have stable spectral patterns and are highly redundant for the pitch classification task. Thus NSynth-Pitch is an ideal use case of DiffRes. (ii) Our statistic shows about 19.7% frames in this dataset are silent frames, thus any dimension reduction rate $\delta$ larger than 19.7% means DiffRes need to remove some non-empty frames to benefit classification accuracy.

| Frames per second / Dimension reduction rate | $25/\delta = 75\%$ | $50/\delta = 50\%$ | $75/\delta = 25\%$ |
|---|---|---|---|
| AvgPool | $90.5 \pm 0.3$ | $91.3 \pm 0.2$ | $92.6 \pm 0.2$ |
| Proposed | $92.1 \pm 0.1$ | $92.4 \pm 0.2$ | $92.6 \pm 0.1$ |

**Table 6:** Experiment result on the pitch classification task on the NSynth dataset. All the experiments use a 10 ms hop size. The baseline performance is $92.5 \pm 0.2$, with 10 ms hop size and 100 FPS.

Table 6 shows our result on the NSynth-Pitch task. All the settings use a dimension reduction rate $\delta > 19.7\%$, which means DiffRes have to remove part of the non-empty frames. If we reduce the temporal dimension with AvgPool, the performance will degrade significantly, while our proposed method can remain similar performance even after reducing 75% temporal dimensions. This result suggests DiffRes not only remove the silent frames but also preserves important frames for classification. The high activeness $\rho$ (see Equation 4) in the non-empty frames shown in Figure 6 and Table 4 also indicate the model has learned to distinguish the importance of the non-empty frames.

## A.2 Automatic Audio Captioning with DiffRes

To further verify the generality of our proposed method, we conduct an extra set of experiments on the automatic audio captioning (AAC) task (Mei et al., 2022b), which can automatically generate natural language descriptions for audio clips. We use the architecture proposed by Mei et al. (2021) and the same DiffRes setting introduced in Section 3 for the experiments. Our experiments are on the AudioCaps dataset. We will try to reduce the input feature size of the AAC task and observe the change in model performance.

We conduct experiments on AudioCaps (Kim et al., 2019), which is the largest public audio captioning dataset with around 50 000 10-second audio clips, and is divided into three splits: training, validation and testing sets. The audio clips are annotated by humans through the Amazon Mechanical Turk (AMT) crowd-sourced platform. Each audio clip in the training sets has a human-annotated caption, while each clip in the validation and test set has five ground-truth captions.

For model evaluation, we use the metrics calculated based on $n$-gram matching ($n$-gram refers to $n$ consecutive words) following previous works (Liu et al., 2022d;b). BLEU$_n$ measures the precision of $n$-gram matching and a sentence-brevity penalty is introduced to penalize short sentences. ROUGE$_l$ calculates an F-measure by considering the longest common subsequence between the candidate and ground truths. METEOR calculates uni-gram precision and recall, taking into account the surface forms, stemmed forms, and meanings of words. CIDEr computes the cosine similarity of weighted $n$-grams between candidates and references. SPICE parses each caption into scene graphs and an F-measure is calculated based on the matching of the graphs. SPIDEr is the average of SPICE and CIDEr and is used as the official ranking metric in DCASE challenge (Mei et al., 2022a).

| $\delta$ ($\approx$FLOPs reduction) | BLEU-1 | BLEU-4 | METEOR | ROUGE$_l$ | CIDEr | SPICE | SPIDEr |
|---|---|---|---|---|---|---|---|
| 0% | 0.658 | 0.235 | **0.232** | 0.473 | 0.643 | 0.168 | 0.406 |
| 25% | 0.665 | 0.247 | 0.228 | 0.471 | **0.657** | **0.171** | **0.414** |
| 50% | **0.674** | **0.266** | 0.230 | 0.475 | 0.646 | 0.167 | 0.407 |
| 75% | 0.659 | 0.252 | 0.225 | **0.478** | 0.649 | 0.164 | 0.407 |

**Table 7:** Applying DiffRes on the automatic audio captioning task, which exhibits a similar trend with audio classification tasks shown in Table 1 and Table 2. By removing 25% of the input dimensions, the performance on some metrics even got improved. After removing 75% of the input temporal dimensions with DiffRes, the model can still retain a comparable result.

The result in Table 7 shows that applying DiffRes on the AAC task can significantly reduce the computational cost while preserving similar performance on most of the metrics. We perform experiments with four different temporal dimension reduction rate settings, including 0%, 25%, 50%, and 75% reductions. The reduction on the temporal dimension also significantly benefits model throughput at the same time (see Figure 5). After removing 25% temporal dimensions, the performance of AAC even shows an improvement, which might be due to the data augmentation effect mentioned in Section 3. After removing 75% of the input temporal dimensions, the model can still achieve on-par results compared with the baseline 0% reduction. The 75% reduction setting can even improve five metrics out of the total seven metrics. The result of the AAC task further indicates our proposed method is generalizable to other similar audio tasks.

## A.3 FIGURES

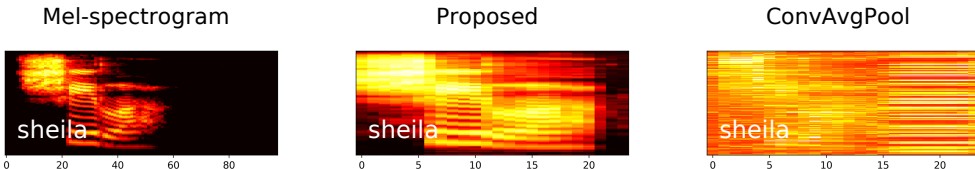

**Figure 9:** Comparison of mel-spectrogram, DiffRes feature, and ConvAvgPool learned feature. The DiffRes feature preserves more details in the original mel-spectrogram and is more interpretable than the ConvAvgPool feature.

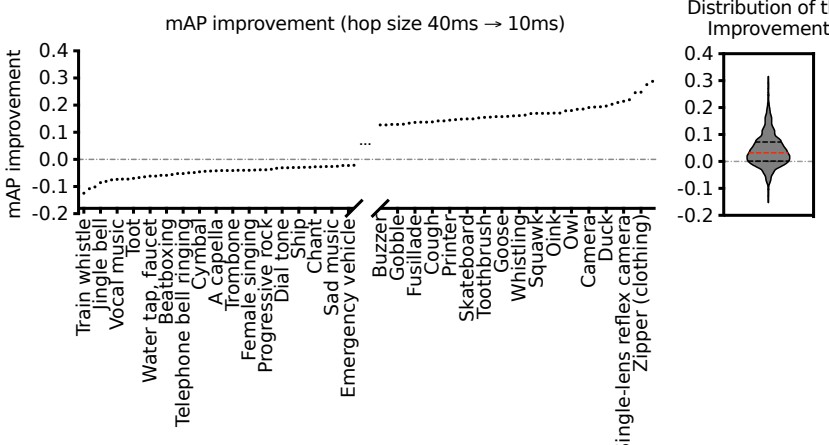

**(a)** Class-wise improvement after changing hop size from 40 ms to 10 ms.

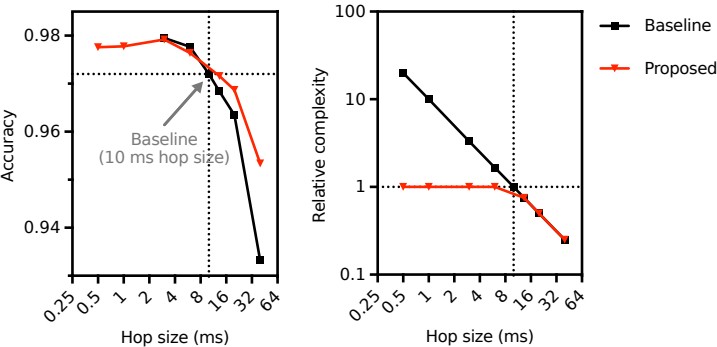

**(b)** Accuracy and the classifier computational complexity with different hop size settings on the speech recognition task. The black dotted lines show the accuracy, and complexity with a 10 ms hop size.

**Figure 10:** Pilot studies. a) The mAP improvement for each class in the AudioSet after decreasing the hop size from 40 ms to 10 ms. The violin plot on the right side shows the improvement distribution, where the red dashed line is the median value. The inconsistency of improvement in different sound classes indicates they need different temporal resolutions to achieve optimal classification performance. a) The accuracy can be improved with a smaller hop size at the cost of computation. DiffRes can achieve similar improvements without increasing computational complexity.

## A.4 WEAKNESS ANALYSIS

Table 2 shows DiffRes does not improve the model performance on 1 ms setting on most datasets. This may be due to the insufficient receptive field of the convolutions in DiffRes, which is around 41 time steps. By comparison, the temporal dimension of $X$ on AudioSet is $t = 3333$ and $t = 10\,000$ with 3 ms and 1 ms hop size, respectively. DiffRes may not effectively capture the useful information with only 41 temporal receptive field in this case. Future work will address this problem by designing the resolution prediction model with a large receptive field.

## A.5 DATASET AND EXPERIMENT DETAILS

| Dataset | Learning rate | Epoch | Batchsize | Learning rate scheduler (start epoch, gamma, every n epoch) | GPU(s) |
|---|---|---|---|---|---|
| Audioset | $1.0 \times 10^{-4}$ | 30 | 22 | $(11, 0.5, 5)$ | 4 |
| FSD50K | $5.0 \times 10^{-4}$ | 40 | 15 | $(21, 0.5, 5)$ | 1 |
| ESC50 | $2.5 \times 10^{-4}$ | 80 | 32 | $(41, 0.95, 1)$ | 1 |
| SpeechCommands | $2.5 \times 10^{-4}$ | 60 | 128 | $(25, 0.9, 1)$ | 1 |
| NSynth | $1.0 \times 10^{-4}$ | 30 | 48 | $(11, 0.85, 1)$ | 1 |

**Table 8:** Hyper-parameter setting. We run all the experiments with an ADAM optimizer (Kingma & Ba, 2014) and GeForce RTX 2080 Ti GPU(s).

| Task | Audio tagging | Audio tagging | Environmental sound | Speech recognition | Music instrument |
|---|---|---|---|---|---|
| Dataset | AudioSet | FSD50K | ESC50 | SpeechCommands | NSynth |
| Classes | 527 | 200 | 50 | 35 | 11 |
| Train examples | 1 912 134 | 36 799 | 2000 | 84 771 | 289 205 |
| Test examples | 18 887 | 10 231 | - | 10 700 | 12 678 |
| Duration (mean,std) | 9.91, 0.50 | 7.63, 7.82 | 5.00, 0.00 | 0.98, 0.07 | 4.00, 0.00 |
| Pad to length | 1000 | 3000 | 500 | 98 | 400 |
| Evaluation metric | mAP | mAP | Accuracy | Accuracy | Accuracy |
| 5-fold cross-validation | - | - | ✓ | - | - |
| Class re-balancing | ✓ | ✓ | - | - | - |
| SpecAug | ✓ | ✓ | ✓ | ✓ | ✓ |

**Table 9:** Detailed information of the datasets we used in this paper. We perform padding to unify the data length. The last row shows the mel-spectrogram temporal dimension we used for batched training.

| | Frame importance estimation module $\mathcal{H}_\phi$ | ConvAvgPool Encoder |
|---|---|---|
| Parameters (k) | 82.4 | 493.8 |
| Kernel size | 5 | 5 |
| ResConv1D blocks (input chnanel, onput chnanel) | $(128, 64), (64, 32), (32, 16), (16, 8), (8, 1)$ | $(128, 128), (128, 128), (128, 128)$ |

**Table 10:** The structure of the frame importance estimation module and the front-end structure of ConvAvgPool (baseline method in Table 1). The structure of ResConv1D is discussed in Section 2.2.1
.

A.6 Algorithm formulation of DiffRes

---

**Algorithm 1:** Inplace warp matrix update

---

**Inputs :** $\boldsymbol{W}$ in Equation 5. $\boldsymbol{W}$ satisfies $\sum_{i=1,j=1}^{t,T} \boldsymbol{W}_{i,j} \leq t$.

**Output:** The updated $\boldsymbol{W}$ that satisfies both $\sum_{i=1}^{t} \boldsymbol{W}_{i,:} = \boldsymbol{1}$ and $\sum_{i=1,j=1}^{t,T} \boldsymbol{W}_{i,j} \leq t$.

1   $i \leftarrow 1; \ j \leftarrow 1; \ s \leftarrow 0$           $\triangleright \ i, j$ for indexing, $s$ store the sum value in each row.
2   **while** $i < t$ **and** $j < T$ **do**
3      **if** $\boldsymbol{W}_{i,j} > 0$ **then**
4         $s \leftarrow s + \boldsymbol{W}_{i,j}$;                $\triangleright$ Add up the weights in row $i$.
5         $j \leftarrow j + 1$;               $\triangleright$ Move to the next column in row $i$.
6      **else**
7         $\boldsymbol{W}_{i,j} \leftarrow 1 - s$;       $\triangleright$ Assign a weight to $\boldsymbol{W}_{i,j}$ to make $\sum \boldsymbol{W}_{i,:} = 1$
8         $\boldsymbol{W}_{i+1,j} \leftarrow \boldsymbol{W}_{i+1,j} - \boldsymbol{W}_{i,j}$;       $\triangleright$ The assigned weight is taken from $\boldsymbol{W}_{i+1,j}$.
9         $i \leftarrow i + 1$;               $\triangleright$ Move to row $i + 1$ in column $j$.
10        $s \leftarrow 0$;            $\triangleright$ Moved to the next row. Reset the row sum value.

---

**Algorithm 2:** Perform audio classification with the DiffRes layer

---

**Inputs :** $\boldsymbol{x} \in \mathbb{R}^L$: One-dimensional waveform. $\mathcal{F}_\phi$: DiffRes layer. $\mathcal{G}_\theta$: Classifier backbone.

**Output:** $\hat{y}$: Classification result.

1   $\boldsymbol{X}^{F \times T} \longleftarrow Q_{l,h}(\boldsymbol{x})$;     $\triangleright$ Project input to the Fourier domain with fixed temporal resolution.
2   $\boldsymbol{O}^{F \times t}, \mathcal{E}^{F \times t} \longleftarrow \mathcal{F}_\phi(\boldsymbol{X})$;   $\triangleright$ DiffRes layer (see Algorithm 3). $\mathcal{E}^{F \times t}$ is the resolution encoding.
3   $\hat{\boldsymbol{y}} \longleftarrow \mathcal{G}_\theta(\boldsymbol{O}, \mathcal{E})$;             $\triangleright$ Classifier forward propagation.
4   **if** *training* **then**
5      $loss = \mathcal{L}_{bce} + \mathcal{L}_{guide}$;       $\triangleright \mathcal{L}_{guide}$ is the loss function for DiffRes (See Equation 7)
6      backprop;

---

**Algorithm 3:** DiffRes layer $\mathcal{F}_\phi$

---

**Inputs :** $\boldsymbol{X} \in \mathbb{R}^{F \times T}$: Fix-time-resolution spectrogram. $\boldsymbol{E} \in \mathbb{R}^{F \times T}$: Positional encoding.
         $\mathcal{H}_\phi$: Frame importance estimation network. $t \in \mathbb{Z}$: Target temporal dimensions.

**Output:** $\boldsymbol{O} \in \mathbb{R}^{F \times t}$: Two-dimensional representation with adaptive temporal resolution.
        $\mathcal{E} \in \mathbb{R}^{F \times t}$: Resolution encoding.

1   $\boldsymbol{s}^{1 \times T} \longleftarrow \text{Norm}(\sigma(\mathcal{H}_\phi(\boldsymbol{X})))$;       $\triangleright$ Calculate frame importance (see Equation 3).
2   $\boldsymbol{W}^{t \times T} \longleftarrow \alpha(\boldsymbol{s})$;       $\triangleright$ Construct a warp matrix (see Equation 5 and Algorithm 1).
3   $\boldsymbol{O}^{F \times t} \longleftarrow \beta(\boldsymbol{X}, \boldsymbol{W})$;       $\triangleright$ Warping frames with the warp matrix (see Equation 6).
4   $\mathcal{E}^{F \times t} \longleftarrow \boldsymbol{E}\boldsymbol{W}^\top$;           $\triangleright$ Encode the information of $\boldsymbol{W}$ into $\mathcal{E}$

---

### A.7 FAST IMPLEMENTATION OF ALGORITHM 1

We provide a pure matrix operation-based version of Algorithm 1 for efficient model optimization on GPUs. Given the output of Equation 5, $W_0$, we will process it into a matrix $W$ with the same shape so that $\sum_{i=1}^{t} W_{i,:} = 1$ and $\sum_{i=1,j=1}^{t,T} W_{i,j} \leq t$. Figure 11 provides an example of the fast implementation discussed in this section.

**Figure 11:** An example of the Algorithm 1 vectorized implementation. The number inside the parentheses means the equation number.

First, we calculate the cumulative sum of the distance between one and the total input weight assigned to each output frame, given by

$$P = \text{cumsum}(1_t - \sum^{T} W_0)1_T^\top, \tag{9}$$

where $1_t$ denotes the all-one column vector with shape $t$. For each row in $W_0$, only the first non-negative element and the first zero after the last non-negative element need update (see Algorithm 1). And we locate those elements by performing the following convolution: convolving (no padding) $M$ with a kernel $K$ of shape $t \times 2$, given by

$$Q = \text{Conv1D}(M, K), \quad K = \begin{bmatrix} -1 & 1 \\ -1 & 1 \\ ... & ... \\ -1 & 1 \end{bmatrix}_{t \times 2}, \quad M = \text{sgn}(W_0), \tag{10}$$

where $\text{sgn}(\cdot)$ stands for a sign function. We pad a column of zero on the first row of the output $Q$. The output $Q \in \{0, 1, -1\}$ is a matrix with shape $t \times T$. Then we calculate the value that needs updates with $P$ and $Q$, given by

$$U = P \odot (-Q)^+, V = \begin{bmatrix} 0_T \\ P_{1:t-1} \odot Q_{2:t}^+ \end{bmatrix} \tag{11}$$

where $P_{1:t-1} = (P_{ij})_{i \in [1:t-1], j \in [1,t]}$ denotes slicing of the matrix row, $0_T$ is the all-zero row vector with length $T$, and $\odot$ denotes element-wise multiplication. Note that here the start index of a matrix is one. $U$ and $V$ store the values that need addition and subtraction for each element in $W_0$. The final warp matrix is calculated by

$$W = U - V + W_0. \tag{12}$$

## A.8 EXAMPLES

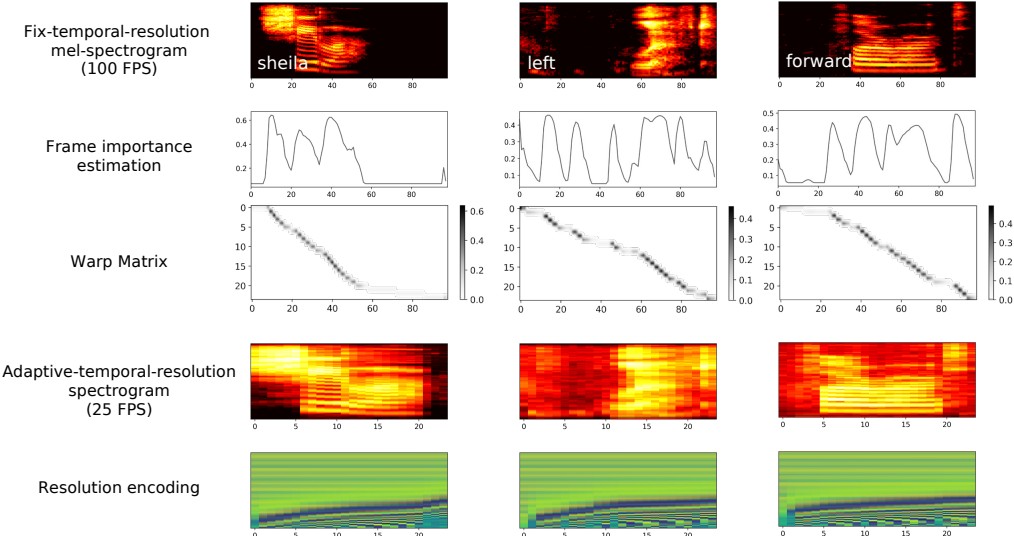

**Figure 12:** Examples of DiffRes adaptive-temporal-resolution spectrogram on the SpeechCommands dataset.

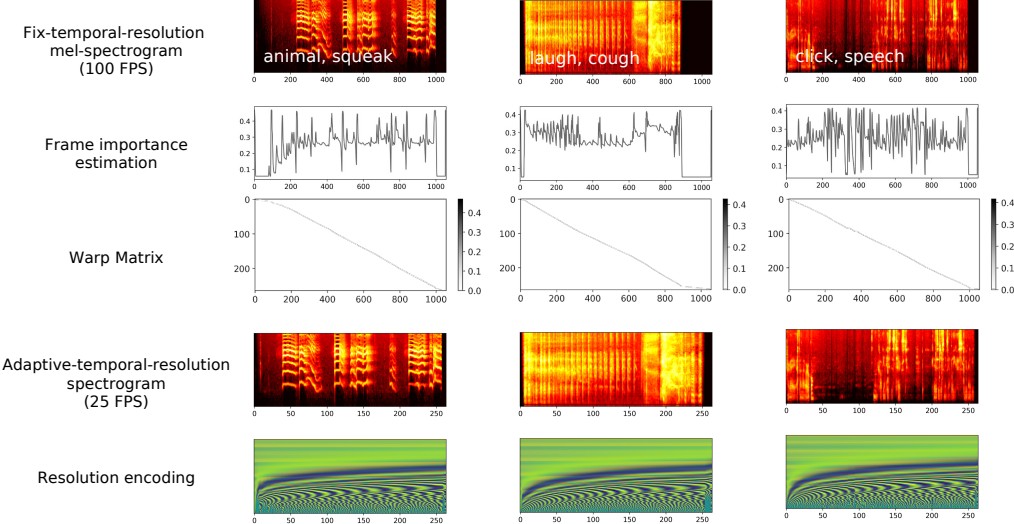

**Figure 13:** Examples of DiffRes adaptive-temporal-resolution spectrogram on the AudioSet dataset.

