# OpenReview forum: "LEARNING THE SPECTROGRAM TEMPORAL RESOLUTION FOR AUDIO CLASSIFICATION"
_ICLR.cc/2023/Conference — Submitted to ICLR 2023_

### Official Review · Reviewer_Uoii · 2022-10-23

**Confidence:** 4
**Correctness:** 3
**Technical Novelty And Significance:** 2
**Empirical Novelty And Significance:** 2
**Recommendation:** 3

**Clarity, Quality, Novelty And Reproducibility:**

The paper is both clear and hard to read. The details of the various steps are clear. However, there are so many engineering tactics that contribute to the overall result that it a challenge to retain focus on the main novelty.  The experimental details, including the listing of the various parameter, along with the availability of the code greatly contributes to reproducibility.

**Strength And Weaknesses:**

Strengths
1. A novel approach to identify the frame rate that is optimal for different audio classification tasks.
2. Code is available online.
3. Experimental details are parameters are clearly stated.
4. Results on several tasks show that the proposed approach results in higher classification accuracy compared to other recent models.

Growth Opportunities
1. The paper makes sweeping novelty claims and misses key references. e.g. Kekre, H. B. et al. “Speaker Identification using Spectrograms of Varying Frame Sizes.” International Journal of Computer Applications 50 (2012): 27-33. and Huzaifah, Muhammad. "Comparison of time-frequency representations for environmental sound classification using convolutional neural networks." arXiv preprint arXiv:1706.07156 (2017).
2. Section 2.2.1 indicates a frame-weighting approach. This is similar to the sample weighting approaches common in ML tasks. Temporal frame warping in 2.2.2 is similar to the use of derived features. These similarities are worth pointing to the reader.
4. The importance of guide loss is understated. It is only in Figure 8 that the large contribution becomes clear. Incorporating that in the abstract, introduction, and conclusion prominently will result in a more faithful representation of the novelty.
5. Ablation study on other neural and event non-neural classification architecture will also be helpful to understand the generalizartion ability of the proposed feature-optimization approach.
6. A lot of experimental tactics, e.g., section 3.1, seem like good engineering recipes. Probably this work is a better fit for the ICASSP or Interspeech community.
7. Legend in table 3 is missing. The difference between the last 2 columns is unclear.
8. Why is the computational cost of LEAF so much higher? An explanation will be helpful.

**Summary Of The Paper:**

The paper proposes approaches to identify the optimal frame rate for audio classification tasks. Specifically, they propose a series of strategies to identify the optimal rate by inserting the frame-rate identification module between the traditional feature extraction step and the final classifier and casting the problem as a joint optimization problem. Experiments on several classification tasks show that the proposed approach improves over recent approaches.

**Summary Of The Review:**

The paper proposes an end-to-end framework to optimize the time resolution and classification accuracy in audio classification tasks. For ICLR, the generality of the approach is not quite established. There are a lot of good engineering tactics though that would be of interest to the audio classification community. I recommend instead audio/speech-based venues such as ICASSP, Interspeech, and MLSP.

---

> ### Author Response · Authors · 2022-11-18
> **Further experiments and discussions are added.**
>
> Dear reviewer Uoii,
>
> Many thanks for the review. Your insight is of great help to improve our paper! We conduct further experiments and make some updates to the paper according to your review (the updated parts are highlighted in red). Hope these explanations are helpful to address the concern mentioned in your review.
>
> **Summary:**
>
> **(1) Further experiments suggest the proposed method** (i) is not sensitive to hyper-parameter in the guide loss; (ii) does not need a lot of experimental tactics; (iii) works in a similar way on other architectures; and (iv) is generalizable to other tasks.
>
> **(2) We update the paper with** (i) an ablation study on hyper parameters and other architectures; (ii) audio captioning result with the proposed method (iii) emphasis on guide loss; (iv) the reason why LEAF (Zeghidour et al., 2021) is much slower; and (v) corrected typos and other references.
>
> **Detailed explanations**
>
> > Ablation study on other neural and event non-neural classification architecture will also be helpful ...
>
> => **We add an ablation study section with other neural architectures in Appendix.A.1.2.** We experiment with two more SOTA audio classification backbones, CNN6 and CNN14 (Kong et al., 2020). Our result shows all three architectures can have a 25% temporal dimension reduction (i.e., approx 25% FLOPs reduction) with the same or better performance. Also, our result shows all three architectures can benefit from DiffRes by learning on a smaller hop size without increasing computational cost.
>
> > A lot of experimental tactics, e.g., section 3.1, seem like good engineering recipes ...
>
> > There are a lot of good engineering tactics ...
>
> => Sorry for the confusion. **The exact setting of a lot of tactics is actually not essential for model training, such as the ratio of spec-augmentation, or mixup rate.** We skip the details of these tactics in the updated version for simplicity. These tactics are widely used in audio classification and will not make our proposed method difficult to use. We report the "tactics" as exhaustive as we can mainly for the reproducibility of our work. **These tactics are controlled to be constant in all the experiments, so our conclusion on DiffRes is not affected by those "tactics".**
>
> **For the hyper-parameters (or tactics) in DiffRes, including delta, lambda, and eps, we provide further ablation studies for them in Appendix. A.1.1.** **Our result shows DiffRes is not sensitive to the choice of those parameters**.
>
> > For ICLR, the generality of the approach is not quite established.
>
> => Firstly, our proposed method can be applied to most of the audio pattern recognition tasks as a tool for removing redundant information. In the paper, we demonstrate our method on multiple datasets and use cases, including multilabel classification~(AudioSet tagging, FSD50K tagging), acoustic scene classification (ESC50), Speech recognition (SpeechCommands), and Music classification (NSynth). We also add the result of DiffRes on the audio captioning task (see next paragraph). **We believe those tasks can already cover a large group of audiences with interest.**
>
> In Appendix. A.2 (new section), to demonstrate the generality of our approach, **we conduct a few more experiments on the automatic audio captioning (AAC) task**. **DiffRes help the AAC model reduce 75% of computations while remaining comparable or better performance.**
>
> Besides, the idea of adaptively merging temporal frames can be useful to other sequence data processing as well, such as video frames. From this perspective, our proposed method is a general way that handles redundancy in sequence modeling by adaptively merging data along one dimension.
>
> > The importance of guide loss is understated. It is only in Figure 8 that the large contribution becomes clear. Incorporating that in the abstract, introduction, and conclusion ...
>
> => We add some sentences in the introduction and conclusion to emphasize the importance of guide loss. We also add one more section (A.1.1) in the Appendix to discuss the effect of different parameters in the guide loss.
>
> > Why is the computational cost of LEAF so much higher? An explanation will be helpful.
>
> => Explanations are added in the main text. For CPU, the majority of computation in LEAF comes from multiple complex-valued convolutions, which are computed in the time domain with large kernels (e.g., 400) and a stride of one.
>
> > The paper makes sweeping novelty claims and misses key references ...
>
> => We add the reference and explanation on page 1.
>
> > Section 2.2.1 indicates a frame-weighting approach. This is similar to ...
>
> => These points are updated in our paper.
>
> >  Legend in table 3 is missing. The difference between the last 2 columns is unclear.
>
> => Explanations are added in the caption.
>
>
>
> **Hope these clarifications are helpful. Looking forward to your reply!**

---

### Official Review · Reviewer_wxoQ · 2022-10-24

**Confidence:** 4
**Correctness:** 4
**Technical Novelty And Significance:** 4
**Empirical Novelty And Significance:** 4
**Recommendation:** 6

**Clarity, Quality, Novelty And Reproducibility:**

Clarity
- This paper is easy to follow. But, there are some parts which were not clear to me.
- (page 5) The guide loss is not straightforward to understand. Why the negative sign was used in front? The epsilon as a hyper-parameter seems to be important as it determines the number of empty frames. I wonder how it affects the overall performance.
- (page 5) The random spec-augmentation was applied directly on mel spectrogram before the diffRes is used.

Quality
- The motivation is clear and the related work is summarized well.
- The technical description is very clear
- The experimental results are convincing

Novelty
- This is the first work that learns temporal resolution on audio spectrogram
- The mathematical formulation is neat

Reproducibility
- The supplementary material includes the source code
- The papers has training details.
- The algorithm is described clearly for implementation


**Strength And Weaknesses:**

Strengths
- The propose module was well motivated in the introduction section with appropriate references
- The technical description is delivered well with the illustrated figure of the computational procedure, visual examples, and detailed algorithm formulation in the appendix section.
- The experiment is comprehensive and rigorous, covering various audio classification tasks and experimental settings
- The ablation study is very convincing and explains the effectiveness of the proposed ideas
- The visual animation in the demo link is very impressive

Weaknesses
- There are some unclear part in writing (See below)
- The performance increment is notable for environmental sounds, whereas it is marginal in speech command and music instrument. This may indicate that DiffRes is more effective when the audio examples has a lot of silent parts. This result could limit the use of DiffRes.


**Summary Of The Paper:**

This paper presents a drop-in differentiable module (DiffRes) that automatically adjusts temporal resolutions of spectrogram input for audio classification.  The DiffRes module computes frame-level importance score and calculate a warp matrix to dynamically scale temporal resolution. The guide loss encourages empty frames to have a low importance score. The DiffRes module was validated with various audio classification tasks including audio tagging, speech command recognition, and musical instrument classification. The experimental results show the effectiveness of the proposed module. In addition, it was compared to other learnable front-end models, showing superior results. The empirical analysis validates the efficiency in computational cost and the effectiveness of the resolution encoding and guide loss.



**Summary Of The Review:**

This paper is a great contribution to audio classification tasks and has a potential to be used for any high-dimensional time series data such as video as the authors suggested in the conclusion.

The proposed DiffRes module is neat and handy. It improves not only the classification accuracy but also reduces the computational cost.

My only concern is whether DiffRes works well only when the audio signals have sufficient empty frames (e.g. environmental sounds) or it would be also effective when the audio signals have consistent energy (e.g. music tracks). I also wonder if DiffRes would be useful for automatic speech recognition.

---

> ### Author Response · Authors · 2022-11-18
> **Further studies and updates**
>
> Dear reviewer wxoQ,
>
> Many thanks for the review. Your positive feedback is much appreciated! We have carefully considered your comments and done further experiments. We also update our paper with two pages of ablation studies in the appendix. Hope our clarifications can help to address your concerns.
>
> > My only concern is whether DiffRes works well only when the audio signals have sufficient empty frames (e.g. environmental sounds) or it ...
>
> **To study if DiffRes only learns to remove silent frames or if it would be also effective when the signal has consistent energy, we design a pitch classification experiment on the NSynth dataset following (Zeghidour et al., 2021).** We will refer to this task as NSynth-Pitch, which is a single-label classification task with 128 classes. We design this task for two reasons: **(i)** The data in the NSynth is mostly instrument sounds, which have stable spectral patterns and are highly redundant for the pitch classification task. Thus NSynth-Pitch is an ideal use case of DiffRes. **(ii)** Our statistic shows about 19.7% of frames in this dataset are silent frames, thus any dimension reduction rate larger than 19.7% means DiffRes need to remove some non-empty frames to benefit classification accuracy.
>
> |            Frames per second            |         25         |         50         |         75         |
> | :-------------------------------------: | :----------------: | :----------------: | :----------------: |
> | **Dimension reduction rate ($\delta$)** | **$\delta$=$75$%** | **$\delta$=$50$%** | **$\delta$=$25$%** |
> |                 AvgPool                 | $90.5$ $\pm$ $0.3$ | $91.3$ $\pm$ $0.2$ | $92.6$ $\pm$ $0.2$ |
> |                Proposed                 | $92.1$ $\pm$ $0.1$ | $92.4$ $\pm$ $0.2$ | $92.6$ $\pm$ $0.1$ |
>
> **Table 6.** Experiment result on the pitch classification task on the NSynth dataset. We report the accuracy of the evaluation set. All the experiments use a $10$ ms hop size. The baseline performance is $92.5\pm0.2$, with $10$ ms hop size and $100$ FPS.
>
> Table 6 shows our result on the NSynth-Pitch. All the settings have $\delta > 19.7\%$, which means DiffRes have to remove part of the non-empty frames. If we reduce the temporal dimension with AvgPool, the performance will degrade significantly, while our proposed method can retain similar performance even after reducing $75$% temporal dimensions. **This result suggests DiffRes not only removes the silent frames but also preserves important frames for classification.**
>
> > The performance increment is notable for environmental sounds, whereas ...
>
> This might because the dataset for environmental sounds (ESC50) is only a small dataset, in which case the data augmentation effect of DiffRes become more prominent. We believe it's not quite practical to compare the absolute improvement values between datasets. Nevertheless, comparing with previouly accepted work (Zeghidour et al., 2021), our improvement on speech and music is already notable.
>
> >  I also wonder if DiffRes would be useful for automatic speech recognition.
>
> We believe "Yes".
>
> 1. From the motivation perspective, features in speech recognition such as MFCC also contains non-informative frames such as noise or silence. Human speech signals are sometimes redundant as well.
>
> 2. In ASR, DiffRes can be a "soft" way to remove non-informative frames. "Soft" means even if the frames are not important, they will be pooled instead of removed (e.g., using voice activity detection). DiffRes is also differentiable, so can be jointly optimized with the down stream tasks such as ASR.
> 3. For streaming ASR, if the importance estimation module becomes causal, the spectrogram can be compressed in real-time, which in turn can support the streaming application.
> 4. Last but not least, DiffRes has already been shown success on SpeechCommands. It would be interesting to see how DiffRes perform on large-scale ASR training. Maybe with more data, DiffRes can better learn the redundancy within different phonemes of human speech.
>
> **We also add one more section for experiments on automatic audio captioning (Appendix. A.2).** Please feel free to checkout.
>
> > - (page 5) The guide loss is not straightforward to understand. Why the ...
>
> Sorry for the confusion. We have updated the formulation so it's easier to understand now. The previous formulation of $\mathbb{S}_{active}$ (Equation 4) is not consistent with our code, and we made some updates.
>
> The minus sign in the guide loss in the previous version is a typo. Apologize for that.
>
> The $\epsilon$ almost does not affect the performance, as discussed in the added Appendix. A.1.1. We also conduct further ablation studies on $\lambda$ and $\delta$ in the guide loss in the Appendix. A.1.1.
>
> > - (page 5) The random spec-augmentation was ...
>
> Yes, we apply the SpecAug before DiffRes. We update the main text to make this clear.
>
>
>
> **I appreciate your time for the review! Looking forward to your reply!**

---

### Official Review · Reviewer_qgYk · 2022-10-24

**Confidence:** 4
**Correctness:** 2
**Technical Novelty And Significance:** 2
**Empirical Novelty And Significance:** 3
**Recommendation:** 6

**Clarity, Quality, Novelty And Reproducibility:**

I find the paper clear and self-contained. The proposed approach is reproducible, and the authors shared the code in an open repository. The novelty of the work is limited, and the proposed approach has substantial limitations. Some of the authors' claims do not agree with the proposed approach, especially concerning temporal resolution learning. The data compression module for spectrogram representations provided by the proposed approach is an interesting idea, but its dependency on several free parameters casts doubts about its generalizability.

**Strength And Weaknesses:**

The paper is interesting and addresses a relevant problem since the information contained in spectrograms is mostly sparse, so data compression can lead to a reduction in the computational load of audio classification systems, such as the authors showed. The authors performed a large enough battery of experiments to show the effects of their proposal on different audio classification tasks and regarding also different hyper-parameters involved in the process.

The main drawback of the proposed approach is the introduction of several free parameters (hyper-parameters) that make the proper training of the module a problematic task. In this sense, the proposed method has a significant limitation regarding the definition of the dimension reduction rate. Such a parameter is similar to any feature reduction rate that must be adjusted following a validation methodology, which goes against the authors' claim that the proposed approach is an end-to-end approach. Following this idea, I also find the title of the paper problematic; the authors claim that the proposed method "learns the temporal resolution," but it is not changing the temporal resolution of the spectrogram; it is applying a compression algorithm to identify the frames with less or not none information at all, in order to reduce the dimensions of the audio representation processed for further modules. The experiments and results presented in section 3 show that temporal resolution must be changed manually, so there is no any gain in this respect by using the proposed module.

On the other hand, the proposed loss function to adjust the s coefficients is quite arbitrary as it is the selection of its parameters. The authors should elaborate more about why it is a good choice for the frame compression task. They should also present results and discussion about the sensitivity of the model to delta, epsilon, and lambda parameters.

**Summary Of The Paper:**

The paper proposes a "drop-in" module and a loss function to compress the information codified in spectrograms for Audio classification tasks. The proposed module is based on standard 1D convolution layers, batch normalization, and residual connections. It can be used as an intermediate module between the input (Spectrogram) and a particular architecture for audio classification. As the authors highlighted, It could also be used with other speech representations either following a feature learning or a feature engineering fashion. The output of the module is a vector containing some frame importance indexes (s_i). Since the proposed module must act as a bottleneck to compress the original information in fewer frames, the proposed loss function promotes the scenarios where the "mass of importance" is distributed in fewer s_i.



**Summary Of The Review:**

The paper proposes a module for spectrogram compression in audio classification tasks. Even though the authors assert that the module learns the temporal resolution, it is not clear how the proposed approach can do that. The proposed formulation lets us see that the module compresses the information of a given Spectrogram, but the temporal resolution is defined during the Spectrogram estimation, so the proposed module cannot do anything about it. Either way, the idea of applying a compression module to Spectrograms is interesting since the information contained in spectrograms is mostly sparse so data compression can lead to a reduction in the computational load of audio classification systems. The estimation of a vector of importance indices is not new since it is the base of the self-attention mechanism. Still, the way such a vector is used to build the matrix W and the corresponding bottleneck representation O is interesting as it follows a less data-driven (more heuristic) but coherent approach. A significant limitation of the proposed approach is the inclusion of many hyper-parameters that could significantly affect the performance of the proposed module.

---

> ### Author Response · Authors · 2022-11-18
> **Clarifications and updates**
>
> Dear reviewer qgYk,
>
> Many thanks for the review. Your suggestions are truly helpful for us to improve our work. We did a few more updates and experiments on the paper. Hope those updates can help to address your concerns.
>
> **Summary:**
>
> 1. Temporal resolution in this paper refers to the resolution controlled by the hop size in STFT [1], instead of the time resolution controlled by the window size. We have updated our paper to avoid this confusion.
> 2. Further experiments (see a new sections in Appendix A.1.1 and Appendix A.2) suggest: the proposed method is not sensitive to $\delta$, $\epsilon$, and $\lambda$ parameters.
>
> **Detailed explainations**:
>
> > The main drawback of the proposed approach is the introduction of several free parameters (hyper-parameters) that make the proper training of the module a problematic task.
>
> > They should also present results and discussion about the sensitivity of the model to delta, epsilon, and lambda parameters.
>
> > ... but its dependency on several free parameters casts doubts about its generalizability.
>
> >  A significant limitation of the proposed approach is the inclusion of many hyper-parameters that could significantly affect the performance of the proposed module.
>
> Thanks for the suggestion. **To address this concern, we add one more section for ablation study in Appendix. A.1.1, in which we study and discuss the effect of $\delta$, $\epsilon$, $\lambda$, and the guide loss.** Please feel free to check out the updated version of our paper (updates are highlighted).
>
> We found $\lambda$ has no notable effect on model final performance and learning activeness. For $\epsilon$, we empirically found it has only a minor effect on the training data. The parameter *dimension reduction rate* $\delta$ is not a parameter for optimization, rather it is a parameter to control the computational cost of the classifier. **These additional results suggest that the proposed method is generalizable and not sensitive to several free parameters.**
>
> **We also run an additional *automatic audio captioning (AAC)* task with the default setting of DiffRes and show improvement both on accuracy and efficiency (see Appendix. A.2).** **DiffRes helps the AAC model reduce 75% of computations while remaining comparable or better performance.** All the evidences indicate that our proposed method is not sensitive to the parameter and is generaliable to new tasks.
>
> > Following this idea, I also find the title of the paper problematic; the authors claim that the proposed method "learns the temporal resolution," but ...
>
> Sorry for the confusion. For the term "time resolution", one of the explanations in the DSP domain is the resolution controlled by the window size of STFT, in which larger windows will have less time resolution but better frequency resolution. **At the same time, in the audio pattern recognition domain, time/temporal resolution can also refer to the resolution controlled by hop size in STFT (Kong et al., 2020).** In this case, a smaller hop size means finer temporal resolution.
>
> We understand that temporal resolution is defined during STFT calculation. But in this paper, the temporal resolution refers to the resolution controlled by the hop size (as mentioned in the first paragraph of the paper). So the resolution can be changed after STFT calculation by frame merging or preserving.
>
> > ... dimension reduction rate. Such a parameter is similar to any feature reduction rate that must be adjusted following a validation methodology, which goes against the authors' claim that the proposed approach is an end-to-end approach.
>
> For clarification, the dimension reduction rate $\delta$ is similar to other parameters such as layer numbers of MLP, which is part of the designed model architecture and is usually not expected to be optimized during training. In practice, the value of $\delta$ should be selected based on the computational limit and is part of DiffRes design, which aims to achieve the best tradeoff between computation (or feature dimension controlled by $\delta$) and the information in the dimension-reduced feature.
>
> Besides, we update our paper and claim our proposed method as *jointly optimized with the classifier*, instead of *end-to-end* optimized.
>
>
>
> **We appreciate your time. Looking forward to your reply.**

---

### Author Response · Authors · 2022-11-18
**Response to reviewers**

We'd like to thank the reviewers for their valuable feedback. The updates to the paper (mostly in the appendix) are summarized as follows:

**Experiments:**

**(i) We have added ablation studies on several hyper-parameters and guide loss in the proposed method, such as the dimension reduction rate $\delta$ and threshold in the guide loss $\lambda$.**  We found $\lambda$ has no notable effect on model final performance and learning activeness. For $\epsilon$, we empirically found it has only a minor effect on the training data. The parameter *dimension reduction rate* $\delta$ is not a parameter for optimization, rather it is a parameter to control the computational cost of the classifier. These additional results suggest that the proposed method is generalizable and not sensitive to several free parameters.

**(ii) We have added ablation studies on different model architectures.** We experiment with two more SOTA audio classification backbones, CNN6 and CNN14 (Kong et al., 2020). Our result shows all three architectures can have a 25% temporal dimension reduction (i.e., approx 25% FLOPs reduction) with the same or better performance. Also, our result shows all three architectures can benefit from DiffRes by learning on a smaller hop size without increasing computational cost.

**(iii) We have studied whether DiffRes only learn to remove the silent frame or if it would be also effective when the signal has consistent energy.** We design a pitch classification experiment on the NSynth dataset following (Zeghidour et al., 2021). The result and discussion in Appendix A.1.3 suggest DiffRes not only removes the silent frames but also preserves important frames for classification.

**(iv) We have tested DiffRes on an additional automatic audio captioning (AAC) task.** We use the default setting of DiffRes in this experiment. AAC shows improvement both on accuracy and efficiency (see Appendix. A.2). DiffRes helps the AAC model to reduce 75% of computations while remaining comparable or better performance. This indicates that our proposed method is not sensitive to the hyper-parameter and can generalize to new tasks.

**Paper:**

**(I) We have updated some equations;**

**(ii) We have corrected a few typos and references;**

**(iii) We have added explanations for a few points that may confuse readers;**

**(iv) We have improved the caption of a few tables.**

---

### Decision · Program_Chairs · 2023-01-20

**Decision:**

Reject

**Justification For Why Not Higher Score:**

Two major concerns.

1. DiffRes shares similarity with some existing techniques such as variable frame rate and sample weighting. Although a neural version of variable temporal resolution is non-trivial and worth investigating, the technical novelty is not overwhelmingly significant.

2.  DiffRes as a drop-in module still needs some hyper-parameter tweaking and engineering tactics in order to obtain good performance when combing with different down-stream tasks in an end-to-end fashion.

**Justification For Why Not Lower Score:**

.N/A

**Metareview: Summary, Strengths And Weaknesses:**

In this paper the authors propose a drop-in differentiable module, DiffRes,  that can learn to adapt temporal resolutions of input spectrograms to jointly optimize with down-stream audio classification tasks. It starts with a small frame hop size, evaluates the importance of the frames and then scales temporal resolution accordingly based on a warping matrix.  A guide loss is also introduced to provide guidance for DiffRes on learning frame importance.   Experiments are carried out on various audio classification tasks and the results are supportive.  The authors show that DiffRes can yield decent performance with a reduction of temporal dimensions and therefore can improve computation efficiency.  It is also possible to improve classification accuracy leveraging high temporal resolution in some task.  In the rebuttal, the authors cleared some of concerns raised by the reviewers and AC with clarification, explanation and added experiments.  All reviewers consider the work interesting and the idea  somewhat novel. The experiments are extensive yet controlled.  However,  there are two major concerns.  First, the proposed DiffRes shares similarity with some existing techniques such as variable frame rate and sample weighting. Although a neural version of variable temporal resolution is non-trivial and worth investigating, the technical novelty is not overwhelmingly significant.  Second,  it seems that DiffRes as a drop-in module still needs some hyper-parameter tweaking and engineering tactics in order to obtain good performance when combing with different down-stream tasks in an end-to-end fashion.